# Exploring experiences of work-related inequitable treatment among international medical graduates (IMGs): A sequential explanatory mixed methods study

Sunita Joann Rebecca Healey[1,2,3]*, Kristy Fakes[1,4], Bunmi Malau-Aduli[1,5], Lucy Leigh[1,6] Balakrishnan R Nair[1,2,3]

1 School of Medicine and Public Health, College of Health, Medicine and Wellbeing, The University of Newcastle, Callaghan, New South Wales, Australia, 2 Equity in Health and Wellbeing Research Program, Hunter Medical Research Institute, New Lambton Heights, New South Wales, Australia, 3 Workplace Based Assessment Program, Hunter New England Local Health District, New Lambton Heights, New South Wales, Australia, 4 Heart and Stroke Research Program, Hunter Medical Research Institute, New Lambton Heights, New South Wales, Australia, 5 College of Medicine and Dentistry, James Cook University, Townsville, Queensland, Australia, 6 Data Sciences, Hunter Medical Research Institute, New Lambton Heights, New South Wales, Australia

* Rebecca.Healey@newcastle.edu.au

## Abstract

### Background

International medical graduates (IMGs) are an essential workforce for many high-income countries worldwide and are often recruited to fill workforce shortages. Studies identify workplace discrimination as a major challenge for IMGs. However, little detailed exploration has been undertaken on this issue.

### Methods

We designed a sequential explanatory mixed methods study to explore details of inequitable treatment perceived by IMGs in Australia. An online survey was distributed to IMGs across Australia. We analysed data descriptively and tested for significant demographic differences against the primary discrimination variable using tests of association (t-test and chi-squared tests). Following analysis, individual interviews were undertaken by telephone, teleconference or face-to-face. Thematic analysis was conducted on qualitative study components. All data was triangulated to assess areas of congruence and difference and to gain fuller understanding of the data.

### Results

We surveyed and interviewed 286 and 36 IMGs respectively. Most survey respondents reported that IMGs were disadvantaged when compared to Australian graduates, primarily due to registration and bureaucratic processes. Institutions/organisations and senior staff were implicated as major perpetrators of discrimination. Subtle experiences of

**Data availability statement:** Ethical restrictions do not allow the public sharing of datasets from this study, due to the risk of directly or indirectly identifying participants from this small and private population. Survey participants have consented to sharing of pooled data only. Therefore, for access to datasets, a variation to the current Ethical approval must be requested and obtained prior to release. The University of Newcastle Human Research Ethics Committee may be contacted via email (human-ethics@newcastle.edu.au). The quantitative data-sets from this study are held in a software database (REDCap) which is hosted by Hunter Medical Research institute, and therefore may be accessed from the Data Management and Health Informatics team at HMRI via email (adminredcap@hmri.org.au). All qualitative data transcript excerpts related to the study are available within the paper.

**Funding:** This research was supported by infrastructure funding from the Hunter Medical Research Institute (HMRI) Equity in Health and Wellbeing Research Program and an Australian Government Research Training (RTP) Scholarship.

**Competing interests:** SJRH and BRN are both IMGs and work closely with IMGs, within the Workplace Based Assessment Program, Hunter New England Health. Other authors (KF, BMA, LL) do not demonstrate any competing interests.

interpersonal discrimination were reported by >75% of those reporting discrimination in the last five years. Statistically significant associations ($p < 0.05$) were identified between the primary discrimination variable and ethnicity, native language, country of primary medical qualification and employment status. Negative sequelae of discrimination on IMG health and career progress were reported. Four themes were identified: i) Disadvantage as an independent construct to discrimination; ii) Structural and institutional discrimination facilitates exploitation; iii) Workplace bullying is a manifestation of inequitable treatment; iv) Inequitable treatment has physical and mental health implications for IMGs. High congruence was detected on triangulation of the quantitative and qualitative results.

## Conclusions

IMGs describe various aspects of discrimination and disadvantage in Australia, warranting further investigation and action. Institutions are responsible for supporting a more equitable and inclusive environment for IMGs.

## Introduction

Workforces globally rely on the migration of skilled migrants to bolster health services, particularly in high income countries [1,2]. Doctors who have obtained their primary medical qualifications (PMQs) external to the country they are based in, are known as 'foreign medical graduates', 'overseas trained doctors' or 'international medical graduates (IMGs)' [3]. We use the latter term for purposes of this article. Worldwide, IMGs are relied upon to fill workforce shortages and are often overrepresented in geographically isolated locations, due to insufficient supply of local medical graduates to service areas of need [4–6].

As institutions seek to safeguard health workforces, there is growing scholarly interest into exploring IMG experiences and mechanisms of support within the workplace [7,8]. Alongside this is recurring evidence that IMGs identify perceived inequitable treatment as a source of challenge in host countries. Motala's 2019 scoping study of IMG challenges identified issues related to workplace discrimination, career limitation and bias in five international studies [9]. Al Haddad's 2021 meta-ethnography exploring IMG pre and post migration experiences observed experiences of racism, marginalisation and discrimination, consequently impacting career progress [10]. In our 2023 scoping review exploring inequitable treatment perceived by IMGs, we identified numerous common reports of inequitable treatment, including inadequate professional recognition, perceived lack of choice and freedoms, personal marginalisation, favouring of locally graduated doctors, verbal insults and perceptions of harsher punishment [11].

Studies exploring racial discrimination toward physicians outside the IMG context are more prominent in the scholarly literature. Results from a questionnaire conducted in Finland in 2019 found discrimination against foreign-born physicians in aspects of management, collegial and patient relationships [12]. Racism has been described as rife in the UK, thwarting career opportunities for doctors of minority ethnicity or foreign graduates [13]. The 2021 UK 'Bridging the Gap' Summary Report described the phenomenon of 'differential attainment'- where career outcomes for doctors are influenced by unchangeable personal factors such as age, gender, race, ethnicity and immigration status rather than ability and effort [14]. The Report explored how structural and social inequalities obstructed professional outcomes and opportunities for doctors of ethnic minority [14]. Discrimination may negatively impact both physical and mental health outcomes by a variety of psychological,

biological and behavioural responses, such that racism can be considered a determinant of health [15,16]. Studies confirm the role of racial discrimination as a negative impact on physician health and career [17].

As IMGs are frequently recruited by host countries to fill workforce shortages, they may be subjected to rules and regulations specific to employment arrangements [6,18]. For example, the '10-year moratorium' is Australia's strategy to bridge workforce shortages, requiring IMGs to work in areas of need for ten years (or less with scaling), subject to billing thresholds [18]. There are also claims that the current registration and licencing examination processes in Australia lack fairness and transparency [19]. Although success in the licencing exams is imperative for most IMGs progressing to registration, exemptions exist for those with PMQs from UK, USA, Canada, Ireland and New Zealand via the 'Competent Authority Pathway' (CAP) [20]. To date, career trajectories of CAP and non-CAP IMGs have not been explored in the literature.

Given the dearth of research in the area, our study aimed to identify and explore experiences of work-related inequitable treatment as reported by IMGs in Australia. We were interested to explore both blatant (explicit) and subtle (implicit) experiences of bias, in the form of prejudice, stereotypes and discrimination as experienced by IMGs. As such, this paper details those personal experiences and perceptions of inequitable treatment, as reported by the study population. The study formed part of a larger PhD body of work, exploring the journeys, and lived experiences of IMGs based in Australia.

## Methods

### Study design and setting

We designed a sequential explanatory mixed methods [21] study to purposefully investigate perceived inequitable experiences of IMGs in two phases. This study design was ideal, as it allowed us to firstly identify broad concepts in the quantitative survey data to then deeply explore through qualitative interviews [21].

The first phase comprised a cross-sectional online survey distributed across Australia between 13 October 2023 and 31 December 2023. The second phase comprised of semi-structured individual interviews, conducted between 6 February 2024 to 18 April 2024; via teleconference, telephone or face-to-face.

### Participants, recruitment and sample size

The survey was conveniently distributed to IMGs Australia-wide. As there was no publicly available contact list for all IMGs in Australia, the researchers used existing professional and personal networks to contact prospective participants, such as social media groups, government organisations, educational/training/support groups; facilitated by snowballing. The names of recruiting networks are deliberately kept anonymous to maintain confidentiality. We chose a range of different networks, aiming to reduce selection bias and improve opportunity to capture a variety of unique journeys and stories.

Participants were eligible to participate if they had obtained their PMQ outside Australia and were living in Australia at the time of survey completion. This included IMGs who were not currently practicing medicine and those in unpaid work or study, to allow those with significant employment impediments to be given an opportunity to express their experiences. As this was an exploratory study, a sample size of 200 was selected as a pragmatic target. This number was deemed feasible to achieve and was expected to provide sufficient variation in demographics and outcomes, enabling the collection of meaningful initial descriptive data from the population.

The interview participants were primarily drawn from the survey respondents who indicated interest in participating, plus a few who directly contacted the primary researcher after hearing of the study through snowballing. We aimed for a minimum of 15 interview participants, with the intention to stop once data saturation was reached, or completion of consent forms returned within the allotted time frame.

## Data collection tools

The cross-sectional survey was designed to gather opinions about a range of IMG-related inequitable experiences. Due to the lack of an existing tool suitable to address our aim and target population, a new tool (see S1 Appendix) was developed, incorporating various components from validated and unvalidated tools from across the literature [11,14,22–26]. For example, we used elements of Nadal's Racial and Ethnic Microaggressions Scale, after adapting it to a five-point Likert Scale [22]. Each survey item was optional, allowing participants to freely skip survey questions and sections, at their discretion. Free commenting was available through open-text options. The remainder of the survey mostly consisted of single choice, multi-choice or Likert scale type responses. The survey was refined by the research team, with collective expertise in both the subject matter and methodology. The instrument was then pre-tested on ten IMGs, for content validation and to assess usability.

We constructed a semi-structured interview questionnaire (see text in S1 Appendix), with intention to explore findings from the survey, as per sequential explanatory mixed methods approach [21]. The study received full Ethics approval with Human Ethics Advisory Panel at the University of Newcastle (H2022-0392) with Access Request approved through Hunter New England Health Human Research Ethics Committee (AR20230405_Nair) and Central Coast Health Human Research Ethics Committee (0323-024C).

## Data collection

Data was collected, stored and managed using the REDCap (Research Electronic Data Capture) software platform, hosted through the Hunter Medical Research Institute [27,28]. REDCap is an online, secure, data capture platform used in health research, providing an intuitive interface for data management and interoperability with common statistical packages [27,28]. The survey and online invitation link were created using REDCap. The online invitation link was forwarded to key contacts of the recruitment networks, who subsequently forwarded the link to individual participants within their network via electronic methods, e.g., email, social media post, newsletter. Each invitation started with information about the research, consent procedures and a QR-link to commence the survey. Commencement of the online survey was explained to participants as implied consent. Data was stored within the REDCap system before being transferred to the statistical program, Stata [29] for analysis.

Interviews were conducted at a time and place convenient to both researcher and participant, by teleconference, telephone or face-to-face, following receipt of written consent. Audio data was collected using data recorders and transcribed by the primary researcher within 48 hours.

## Analysis

Quantitative data: The data was analysed descriptively using Stata (17) [29]. We were interested in understanding the demographic characteristics of the sample and identifying if there were any differences between common demographics and reports of discrimination in the last five years.

Some variables were re-coded or combined for better fit or small response numbers. For example, twelve Ethnicity categories (including open response 'other') were provided as options. Any participant who marked ≥ 1 category were reallocated to a 'mixed' group.

'Other' were reallocated to more appropriate categories if identified (E.g., 'Sri Lanka' was reallocated to the broader 'Asian' category). Therefore, 'Asian' ethnicity was a broad option, including participants identifying as Sri Lankan, Afghani, Filipino, Malaysian etc amongst others who identified as 'Asian'. A distinct ethnic group- 'Asian (Indian)' was provided as an option to participants, reflecting India's significant history as a major exporter of doctors worldwide [30]. Regarding specific experiences of discrimination, responses were recoded into 'any report of discrimination' [(1) 'a little/rarely' (2) 'sometimes/a moderate amount' (3) 'often/frequently'], versus 'no report of discrimination' [(0) 'never'] in the last five years. Similarly, in relation to professional and systemic challenges, [(1) 'strongly disagree' (2) 'slightly disagree' (3) 'neutral' (4) 'slightly agree' (5) 'slightly disagree')] were re-grouped for reporting to three broader categories of 'disagree', 'neutral', 'agree'.

The primary discrimination tool was the binary question: *"In the last five years, have you ever felt discriminated working/attempting to work as an IMG in Australia?"* Responses were assessed against several demographic and pertinent study variables. We used t-tests to assess whether age differed across levels of the primary discrimination tool, and chi-squared tests to test associations between discrimination and categorical variables.

Qualitative data: With the aid of NVivo [31], we undertook thematic analysis on the open-ended responses, guided by Braun and Clarke's principles [32]. Two researchers co-created a coding tree, which was then used by the primary researcher (SJRH) to code and develop themes. The process was overseen by the secondary researcher (KF). We later separately undertook thematic analysis on the interview data, in a similar fashion, with added co-coding of 10% (four) transcripts by both researchers, before the remaining transcripts were individually coded by the primary researcher (SJRH). Emerging themes were discussed regularly. After finalising the themes, the research team conducted a comparative analysis of the themes, derived from the survey open-ended responses and the interview data. We evaluated the alignment between the two sources and integrated themes where appropriate. A third researcher (BMA) reviewed and confirmed identified themes.

Triangulation of data from quantitative and qualitative sources: The full analysis of quantitative and qualitative data was presented on several worksheets which the primary researcher (SJRH) had used to connect, compare and contrast findings. The process was assisted and overseen by an expert in methodology (BMA). We used the Good Reporting of a Mixed Methods Study checklist [33] to report our results; see data in S2 checklist.

## Stance and reflexivity

We used a stance of pragmatism for this study. The major underpinning of pragmatist epistemology is that knowledge can be explored and constructed employing both subjective and objective information. The primary researcher who interviewed the participants (SJRH) is an IMG and a person of colour, with 20 years clinical work experience in rural and metropolitan areas across four Australian states/territories. To reduce the risk of bias, SJRH kept a journalling log and met regularly with KF, a non-medical researcher to debrief, discuss findings and minimise researcher bias during the process. Several interested participants were excluded from participation in the interviews, as they were closely known to the interviewer. This was done to maintain distance and objective stance in the study's data collection and interpretation of findings.

## Results

### Demographics

**Survey participants.** A total of 286 IMGs participated in the survey, with a variable number completing each section due to the optional nature of the survey items (data in

S3 Table.) Survey participants came from diverse demographic and training backgrounds, arriving in Australia with PMQs from a total of 46 countries. Overall, 23.5% (54/230) participants graduated with PMQs from CAP countries, i.e., UK, Ireland, USA, Canada or NZ; therefore, the majority (176/230; 76.6%) graduated from countries outside this pathway. Over half of the survey participants (55.6%; 135/243) reported being of Indian or Asian ethnic background. Forty-four native languages were reported; the top three being English (61/241; 25.3%), Hindi (19/241; 7.9%) and Arabic (16/241; 6.6%). The majority of survey participants reported being part of a religious group: most commonly Christianity/Catholicism (72/239; 30.1%), Hinduism (45/239; 18.8%) or Islam (40/239; 16.7%) and a sizeable proportion (52/239; 21.8%) of survey participants were attributed to a non-religious group, identifying themselves as being agnostic/atheist, non-religious or secular. Most survey participants (190/213; 89.2%) reported working in clinical/specialty domains. Seventeen specialties were represented, including General Practice, Emergency Medicine, Physician/ General Medicine, Paediatrics, Surgery, Anaesthesia, Psychiatry, Obstetrics and Gynaecology, Public Health, Radiology, amongst others. Non-clinical IMGs were represented (24/213; 11.3%), as were a small number (11/213; 5.2%) of IMGs working in fields unrelated to Medicine. Eleven participants worked across a combination of work-settings.

**Interview participants.** Interview participants were primarily drawn from the survey participant pool, resulting in comparable demographic characteristics between survey and interview participants, as detailed in S3 Table. The 36 interview participants had PMQs from 16 different countries. Most (23/36; 63.9%) interview participants self-identified as being a person of colour, or non-Caucasian; the remaining (13/36; 36.1%) participants self-identified as Caucasian. Most (29/36; 80.6%) reported being fluent in two or more languages. Several (7/36; 19.4%) participants had also completed higher qualifications such as Diplomas, Masters and PhDs. Several (11/36; 30.6%) participants had worked in at least two other countries prior to arriving in Australia, and two participants reported working in Australia for a second time period. Several (13/36; 36.1%) participants reported working in more than one state since being in Australia, including four participants who had previously worked in the Northern Territory and/or in South Australia. Participants reported currently working in a range of specialties, including General Practice, Medicine, Surgery, Anaesthesia, Emergency Department, Paediatrics, Obstetrics and Gynaecology, Psychiatry and Intensive Care. Two participants were also employed in non-clinical roles. The majority of participants were interviewed by telephone (21/36; 58.3%); others by teleconference (e.g., Zoom 14/36; 38.9%); and one participant was interviewed face-to-face.

## Quantitative results

**IMGs and reports of disadvantage (see S4 Fig).** According to a five-point Likert scale, when *asked "Overall, when compared to a local Australian graduate, do you think that being an IMG is an advantage or disadvantage?"* The majority of respondents (181/208; 87.0%) reported that IMGs are very (103/208; 49.5%) or slightly (78/208; 37.5%) disadvantaged when compared to local graduates. A minority (9/208; 4.3%) reported that IMGs are slightly or very advantaged when compared to local graduates. The highest reason for respondents to report disadvantage was registration/ bureaucratic requirements (125/181; 69.1%), followed by the way staff treat IMGs (88/181; 48.6%) and assessment requirements (86/181; 47.5%); see data in S4 Fig. Of the nine participants who reported that IMGs were advantaged, the most common reasons were registration/bureaucracy (4/9) and assessment requirements (4/9).

**IMGs and reports of discrimination.** Over half the respondents had responded, 'yes' to *"Since migrating, have you ever felt discriminated working/attempting to work as a doctor in*

*Australia?"* (133/208; 63.9%) and *"In the last 5 years, have you ever felt discriminated working as an IMG in Australia?"* (121/204; 59.3%).

The most reported reasons identified for discrimination were IMG status holding a foreign degree (97/143; 67.8%), followed by language/accent (67/143; 46.9%), race or skin colour (66/143; 46.2%), culture (39/143; 27.3%) gender (29/143; 20.3%) and immigration status or nationality (25/143; 17.5%). Other reasons included, religion, name, age, sexual orientation, marital status and other non-specified reasons. Sixty-seven participants reported three or more reasons why they felt they had been discriminated against in the last five years. Perpetrating sources of discrimination, as identified by affected respondents are shown in data in S5 Fig. Institutions/organisations (the "system") (78/140; 55.7%), and senior staff members or bosses (69/140; 49.3%), were most commonly reported, followed by patients or their families (53/140; 37.9%), medical colleagues (47/140;33.6%), nurses or allied health staff (44/140; 31.4%), other staff, e.g., administration, security officers etc (20/140; 14.3%) and other (6/140; 4.3%) (S5 Fig). Over half of the respondents 90/140 (64.3%) reported two or more perpetrating sources of discrimination.

**IMGs and specific experiences of discrimination in the last five years.** After indicating experiences of discrimination in the last five years, 187 participants were redirected to answer specific questions about their personal experiences (See S1 Appendix). Completion rates on these items ranged from 61.0%-64.0% (114-120/187). This section asked participants to rate their experiences on a four-point Likert scale (never, a little/rarely, sometimes/a moderate amount, often/frequently) regarding examples of explicit and subtle discrimination; and a five-point Likert scale (strongly disagree to strongly agree) regarding examples of discrimination in professional situations and systemic discrimination. Reports of discrimination are tabulated (see data in S6 Table), with calculated dichotomised columns indicating either 'no'/ 'disagree': no experience of discrimination, or 'yes'/ 'agree': any experience of discrimination.

Subtle experiences of discrimination such as feelings of exclusion, assumptions, being treated as less intelligent or inferior, or being treated with suspicion or rudely were reported by over 75% of section respondents. The most common explicit experience of discrimination was experiencing derogatory comments, gestures or teasing (102/120; 85%). Patient refusal, being told to 'return to home country' and unfair complaints were all rated similarly (~40% respondents); see data in S6 Table

Many respondents who reported discrimination in the last 5 years reported challenges with professional and system-related experiences. Most reported was the notion of needing to work 'double hard' when compared to a local graduate (98/118; 83.1%), followed by colleagues with less experience progressing further (83/117; 70.9%) and professional experience/qualifications being questioned or challenged (75/118; 63.5%). System experiences such as limitations of choice in relation to geographical location, limited opportunities in job acquisition, and limited opportunities in career progression were similarly reported (~70% respondents); see data in S6 Table.

**Factors associated with discrimination reported in the last five years (see Table 1 below).** 'English as a native language', 'ethnicity', 'country of PMQ' and 'employment status', were all statistically significantly associated with the primary discrimination tool (p < 0.05). Respondents who reported a native language other than English were more likely to report discrimination in the last five years [94/148; 63.51% vs 27/56; 48.2%; p = 0.047]. A third of respondents (9/25;36%) identifying as British/Irish ethnicity reported discrimination in the last 5 years, compared to half (13/23; 56.52%) European/ Eastern European respondents, and two thirds (91/140; 65%) respondents from all other ethnicities; p = 0.023. Respondents who had PMQs outside the CAP countries were more

**Table 1. Categorical variables associated with discrimination reported in the last five years.**

| Category | | | Discrimination experiences in the last 5 years | | |
|---|---|---|---|---|---|
| | Variable | N | Yes (%) | No (%) | p-value |
| Gender | Female | 147 | 90 (61.22) | 57 (38.78) | 0.390 |
| | Male | 55 | 30 (54.55) | 25 (45.45) | |
| Marital status | Married or de facto | 157 | 95 (60.51) | 62 (39.49) | 0.998 |
| | Non-married/de-facto (e.g., single, divorced, separated, widowed) | 38 | 23 (60.53) | 15 (39.47) | |
| Ethnicity* | British/Irish | 25 | 9 (36.00) | 16 (64.00) | **0.023** |
| | European/ Eastern European | 23 | 13 (56.52) | 10 (43.48) | |
| | All other | 140 | 91 (65.00) | 49 (35.00) | |
| Native language* | English | 56 | 27 (48.21) | 29 (51.79) | **0.047** |
| | Other | 148 | 94 (63.51) | 54 (36.49) | |
| Religion | No religion | 47 | 24 (51.06) | 23 (48.94) | 0.199 |
| | Religion | 141 | 87 (61.7) | 54 (38.30) | |
| PMQ country* | Competent authority pathway country | 51 | 23 (45.10) | 28 (54.90) | **0.017** |
| | All other countries | 148 | 95 (64.19) | 53 (35.81) | |
| Employed* | Working part time or full-time | 184 | 114 (61.96) | 70 (38.04) | **0.034** |
| | extended leave, volunteer, retired or disability pension | 19 | 7 (36.84) | 12 (63.16) | |
| Training | Yes | 43 | 30 (69.77) | 13 (30.23) | 0.122 |
| | No | 157 | 89 (56.69) | 68 (43.31) | |
| Work region | Rural, remote or mixed | 62 | 41 (66.13) | 21 (33.87) | 0.205 |
| | Metropolitan only | 124 | 70 (56.45) | 54 (43.55) | |
| Registration | Full | 117 | 64 (54.70) | 53 (45.30) | 0.105 |
| | Partial, limited or non-practicing | 59 | 42 (71.17) | 17 (28.81) | |
| | None | 24 | 15 (62.50) | 9 (37.50) | |

*indicates significant p-value associated with this category.

likely to report discrimination in the last 5 years [23/51; 45.1% vs 95/148; 64.2%; p = 0.017]. Working respondents were more likely than those on leave/extended breaks to report discrimination [114/184; 62.0% vs 7/19; 36.8%; p = 0.034]. The primary measure of perceived discrimination within the last five years demonstrated no statistically significant association with age [mean age of respondents answering 'yes': 41.18 (95% CI: 39.63 – 42.73); mean age of respondents answering 'no': 40.28 (95% CI: 38.15 – 42.41); p = 0.487], nor the categorical variables of 'gender', 'marital status', 'religion', 'training', 'registration status' and 'work region': see Table 1 below.

**Effects of discrimination on career and health (see** S7 Fig**).** One third of respondents (68/205; 33.2%) reported that workplace discrimination in the last five years had affected their career progression or attainment of higher work positions. A sizeable proportion of respondents also reported effects on their physical and mental health or wellbeing; 40/198 (20.2%) and 97/203 (47.8%) respectively; see data in S7 Fig.

## Qualitative results

We identified four major themes related to perceptions of inequitable treatment by IMGs: i) Disadvantage as an independent construct to discrimination ii) Structural and institutional discrimination facilitates exploitation; iii) Workplace bullying is a manifestation of inequitable treatment; iv) Inequitable treatment has physical and mental health implications for IMGs.

### Disadvantage as an independent construct to discrimination

a) Disadvantage

"… it *does* matter that you look the part, and you understand the doctor-patient culture, you speak in a way that's comprehensible…So…are they disadvantaged? Yes. But is it always with some horrible malintent? No, I don't think so." [Interviewee #21; PMQ: UK]

Some interviewees reported that disadvantage was an expected outcome of being a foreigner in a new land, with different communication styles, skills, and culture. IMGs reported that the familiarisation of Australian graduates with the system and culture naturally gave an expected "edge" in gaining immediate trust and rapport with staff and also being efficient in the workplace. Interviewees also reported that Australian graduates benefited from opportunities to engage in research, form local collaborations, cultivate and build professional relationships throughout their undergraduate training, particularly for those who attended university locally. Many interviewees reported that Australian graduates and native English speakers were also advantaged by their superior command of English, familiarity with colloquialisms and also carried skills to better present themselves during job interviews.

Interviewees and survey open answer respondents reported other disadvantages when compared to Australian graduates. By needing to clear Australian Medical Council (AMC) assessments and/or gain permanent residency status prior to progressing through training, non-CAP IMGs were delayed in career progression. Having dependents/family was reported to impact on IMG capacity to socialise outside the workplace, thereby impairing informal networking and possibly careers progress.

Many interviewees reported that their own culturally normative behaviour in the workplace inadvertently disadvantaged them in Australia. For example, IMGs identified that they might present as: "submissive…not confident in speaking up" (Interviewee #10; PMQ: India) or "overly direct" (Interviewee #31; PMQ: Germany). One interviewee from Trinidad and Tobago explained that their relaxed and 'stress-free' demeanour was misinterpreted by senior staff as disinterest in work. The interviewee continued to explain that "…[my relaxed demeanour]…. worked against my favour. And that's just how I am, because of where I'm from. Everyone from where I'm from, is exactly the same" [Interviewee #32; Trinidad and Tobago]. Interviewees reported that cultural presentations were misinterpreted by staff and patients. Some interviewees reported how showcasing their merits, success and skills were culturally frowned upon in their homeland but celebrated in Australia. One interviewee explained: "… coming from Asia particularly, puts you … downplay. You taught- be humble. Don't brag. But in this country [Australia], [it] is the other way around. Like if you're good, you talk about it, you tell it" [Interviewee #14; PMQ: China].

Some interviewees reported how multiple relocations disadvantaged their family stability and career choices. One interviewee reported that their child had moved 6 schools in 6 years due to work opportunities. Recurrent relocation was also expensive and destabilising, reducing opportunities for IMGs to form solid working and social relationships in any particular location.

b) Discrimination

"I was treated different because I was different. I wasn't white. I wasn't an Australian graduate" [Interviewee #32; PMQ: Trinidad and Tobago].

Separate to disadvantage, were reports of clear discrimination, where IMGs reported unfair preferences exhibited toward local graduates or those of Caucasian background. Other reports included significant mistreatment, exclusions and bullying directed only toward IMGs, or IMGs of certain PMQ backgrounds. Several interviewees reported being disrespected and treated as an "inferior doctor" or "second degree" doctor, by particular personnel at institutions.

Many interviewees described unfair preferential bias toward local graduates, particularly in the way senior doctors supported them. Several interviewees and survey open answer respondents claimed "nepotism" in Australian training programs and academic institutions, where locals were heavily favoured and promoted: "who you know, rather than merit" [Interviewee #31; PMQ: Germany]. A few interviewees also noted that this phenomenon was not necessarily isolated to IMGs, but also to Australian graduates from interstate, or simply those who had not trained in the local health system. Several interviewees told how local graduates were given preference and protection or safe-guarded by senior doctors to practice procedural skills, hot cases and theatre time. Interviewees told how senior staff and rostering showed differential treatment to local graduates compared to IMGs, even those in speciality schemes. Three interviewees left their specialty training schemes because they felt differential treatment and a lack of support.

Observations of differential treatment were common, one IMG writing: "It is confronting to see what permits 'Australian trainees' can enjoy" [open ended survey response]. The waiving of AMC assessment processes for CAP IMGs was identified by IMGs as a manifestation of systemic discrimination; and the propensity for bias in assessments was also reported: "Racist attitude prevails when examiner could see the colour of the applicant!" [open ended survey response]. Furthermore, discrimination was described as a broader problem within society. One IMG wrote: "There's a reason the new consultants advertise themselves as 'Australian graduate'. It's an advantage" [open ended survey response]. Speaking with a foreign accent was described by some interviewees as a trigger for discriminatory treatment received from staff and patients/families over the telephone, and also in-person.

Interviewees also reported feeling overly scrutinized, needing to perform at "double to thrice the amount of work and work ethic" [Interviewee #07; PMQ: Nigeria] in order to reach the same status as their local graduate peers, due to perceived implicit biases around trust coupled with other disadvantages. One interviewee explained: "… [as] an overseas trained doctor …you are assumed incompetent, and you have to prove your competence. Whereas if you're a local graduate you're assumed competent, and then you just have to do the bare minimum" [Interviewee #11; PMQ: India]. Some interviewees reported that this scrutiny (especially from nursing staff) dissipated over time, as they 'proved their worth' and gained trust and respect. Some interviewees told how their mistakes were not easily forgiven, compared to local graduates.

**Structural and institutional discrimination facilitates exploitation**

"We were so called 'visa doctors' so if they say jump, we had to jump" [Interviewee #11; PMQ: India].

Many IMGs reported being disadvantaged by structural discrimination in Australia. By virtue of the workforce being employed primarily to fill gaps in areas of need, many interviewees and survey open ended respondents reported receiving unfair allocations of unfavourable shifts, rotations and geographical restrictions which impacted opportunities for career and family. Many IMGs (including those from CAP countries) reported that the dichotomised CAP versus non-CAP registration pathways were unfair and that current terminology insinuated that doctors graduating from non-CAP PMQ countries were 'non-competent'.

Furthermore, some interviewees and survey open ended respondents described the AMC exam process and other external courses as "fleecing money" from IMGs, with low pass rates, lack of transparency and lack of supportive educational resources. Interviewees explained that exemptions from AMC assessments allowed CAP IMGs superior opportunities to settle more readily in Australia and advance more freely through jobs, residency and prospective training positions. The high load of assessment and examination requirements, coupled with frequent relocations between regional towns reportedly limited the time IMGs could invest in other

post-graduate training activities. Interviewees explained that frequent relocations made it difficult for IMGs to form roots and develop a repertoire of solid referees.

Rural placements were also seen as a disadvantage for training, due to lack of exposure to complex cases, thereby disadvantaging training and chances of success at fellowship examinations. Interviewees described that local graduates received priority for highly sought after mandatory terms as hospital residents, which resulted in a high number of IMGs stagnating in junior positions. One IMG observed: "I've seen how other IMGs have been, 'used', if we can use that word, to stay in rotations and they don't progress and don't get general registration" [Interviewee #05; PMQ: Sweden]. Some IMGs reported unfair shares of weekend, night shifts or relief work. Interviewees mentioned how unfair and excessive allocations were both destabilising and isolating, particularly for newly arrived IMGs. Such shifts also disadvantaged IMG's capacity to study due to fatigue and unavailability to attend tutorials.

IMGs reported being allocated to positions which were not matched to their experience, subsequently promoting de-skilling, or being paid less than their skill-set services deserved. IMGs reported that institutions took advantage of expertise of IMG experiences without acceptable compensation. One IMG told of a colleague who was being paid as an intern, although employed as an unaccredited registrar in a speciality of which they had over ten years overseas experience. IMGs described being more likely to accept unfair allocations as not to risk their visa status, thereby perpetuating their susceptibility to unfair shift work allocation: "I have noticed IMGs will accept unfair or illegal rostering/allocations because we are subconsciously made to feel 'less than' and that we should be 'grateful for the work opportunity'" [open ended survey response]. IMGs reported feeling powerless to ask for what was fair: "…you're afraid to speak… So, you sort of put up with a lot of things... [if you] make a noise…you're not going to get far… and you're risking your whole livelihood" [Interviewee #10; PMQ: India].

Many IMGs reported that the workplace sponsored visa program facilitated exploitation by employers toward IMGs. IMGs used terms such as "shackle", "ransom", "exploit" and "enslave" when describing their relationship with institutions. Two IMGs reported unexpectedly working night shifts for two years. IMGs reported that the current pressures in General Practice (GP) disproportionately affected GP IMGs, leading to burnout. The two-tiered Medicare billing systems which disadvantaged IMGs under moratorium restrictions was described as discriminatory. Furthermore, individual exploitation by practice owners and institutions was reported by various IMGs and considered to be facilitated by current government immigration policies. IMGs reported how practice owners and institutions would control billing, threaten to revoke visas, or withhold pay and contracted leave entitlements from IMGs. This was particularly problematic for IMGs who had relocated their families from developing countries, with intention to gain permanent residency. IMGs told how employers held power over IMGs, as their provider numbers were linked to the practice, which was "shackled" to their visa. One IMG was told of unfair work demands, where IMG GPs were told by practice owners "… to be here 24 hours a day, and then you know, sleep on the sofa, and if anybody [patient] comes in overnight, then you have to see them" [Interviewee #13; PMQ: UK]. IMGs reported that threats of visa revocation or sabotage to their career made IMGs "very scared to complain".

**Workplace bullying is a manifestation of inequitable treatment**
"Dare I say, the darker the skin, the worse the outcome" [Interviewee #27; PMQ South Africa]

Several interviewees described receiving or observing bullying in the workplace. Openly offensive or discriminatory remarks were described, as well as a sense of general rudeness and hostility received from staff and/or patients.

Interviewees reported snide, demoralising or offensive remarks directed to the quality of their medical school or background education, religious background and/or physical

appearance. An Egyptian IMG was told by a colleague that he "looked like a terrorist". A Brazilian IMG reported medical colleagues referring to her as "a sexy Latino", making her extremely self-conscious about her appearance in the workplace. One UK IMG recalled her surprise of racism in Australia: "I remember one orthopaedic surgeon talked about one study that 'the Japs' did. My head of department talked about …Arabic people as 'ragheads'. And this was to me…people would be talking about some kind of South Asian racial stereotype to *me*!" [Interviewee #12; PMQ: UK]. One Muslim IMG reported that he was unfairly pre-judged by staff about attitudes toward women in the workplace, and was accused of lying, being told that " lying is normal from where *you* come from, [but] lying is not normal in this part of the world" [Interviewee #20; PMQ: Egypt]. Female IMGs reported an added element of discrimination- one female Indian IMG was openly accused of not working hard enough at a departmental meeting. One German IMG was asked to consult her husband before increasing her clinical hours.

Interviewees reported that major perpetrators of personal mistreatment were seniors/consultants, nursing staff and less commonly, patients. Several IMGs reported that among the worst offenders were fellow IMGs who held senior positions; one IMG described her surprise: "…they just treat you very bad, very bad. Not all consultants are like that, but many are…. some were Australian, and some were not. That is what left me speechless sometimes. Like, *you are an IMG like me. Yes, you came here, maybe 20 years ago, but how can you treat me like that? You know what it's like!*" [Interviewee #19; PMQ: Italy]. Several IMGs observed the way IMG consultants responded to non-Caucasian IMGs was different to the way they responded to local graduates, or Caucasian IMGs. Many theories of this phenomenon were postulated by interviewees, including subconscious cultural reactions, reasserting a familiar hierarchical relationship, internalised racism, power play and simply an opportunity to 'get away with it'. One UK IMG observed: "For some reason, they wouldn't behave like that to anybody else. But they would to their own. Because I think back home, that is the way it is. You're like, '*Yes sir! No sir!*' Your Consultant's God" [Interviewee #21; PMQ: UK]. Interviewees described the elevated societal position and cultural respect afforded to doctors in the context of their home countries. Some interviewees described cultural taboos and lack of confidence in standing up against those in authoritative positions which therefore perpetuated bullying behaviour in Australia.

Although a few interviewees accepted that their accents were strong, all interviewed participants were confident with their English language skills and cited success in English proficiency tests as validation. However, some IMGs reported that complaints raised by staff or patients regarding difficulties in understanding accent were in fact, an excuse for underlying hidden racism. Native English-speaking IMGs from the UK and Trinidad and Tobago, with non-Caucasian appearances had been told that their English was poor.

Sometimes the hostility described by interviewees was subtle and covert: "They don't tell you, I discriminated against you, but you feel it, you can see it in their eyes, in the way they deal with you. They don't say they are racist, but they do racist things, discriminatory things" [Interviewee #20; PMQ: Egypt]. Another IMG reported that she felt people kept physical distance from her because of her religious scarf. One IMG observed patients being racist, saying: "I feel like Australia is even more racist than South Africa in some ways. It probably just doesn't get the label…" [Interviewee #27; PMQ: South Africa]. A few interviewees reported that staff were impatient and unfriendly, openly showing their annoyance of IMGs (e.g., rolling eyes), or degrading them publicly. An undercurrent of societal racism and subconscious biases and prejudices was postulated by several IMGs as the driver of interpersonal hostility.

Some interviewees reported the extra scrutiny of IMGs resulted in unfair complaints and workplace gossip. Only a few interviewees reported escalating their concerns past

their workplace to advocacy groups such as the Australian Medical Association, medical indemnity, human rights or fair work groups. Several reported that they were not aware of the procedure to voice complaints or how to obtain independent advocacy. One IMG described that her complaints against other staff members were gaslighted: "I couldn't voice my opinion. My voices were not heard. And when I said that 'this senior is doing this to me…' no one would side …they would say - 'I'm really concerned about the way that you are speaking about the people who is [are] trying to support you'" [Interviewee #35; PMQ: Ireland].

**Inequitable treatment has physical and mental health implications for IMGs**

"She actually went to the extent to tell me that she can sabotage my…[training] pathway…I was planning to quit the job, and I was crying all the day… Even when I'm talking to you, I have these goosebumps…" [Interviewee #09; PMQ: India].

Some interviewees reported that unfair treatment at work or discriminatory institutional set-ups led them to prematurely resign from jobs or training, move interstate, change careers or consider leaving Australia. However, many more reported various effects of inequitable experiences on their health.

Mental health effects included: mood disorders, anhedonia, stress, fear, low confidence and feeling de-valued. Discriminatory comments affected IMG perceptions of self-worth: "It makes you feel like…inferior, and stupid and idiot… because you've got an accent…" [Interviewee #19; PMQ: Italy]. Many interviewees reported that their self-esteem and sense of accomplishment were impacted because their experiences and expertise were not acknowledged by institutions. The psychological and financial pressure of the AMC examinations were considered unfair and damaging. Interviewees described how compounding stressors affected their mental health: " … in a term, I had worked 11 out of 13 weekends. And then I have worked Nights, Nights, Nights… So that has affected my mental health and physical health…I was depressed…It's a mix of scenarios, not just working, but exams as well…[and] your migration status, like every year you have to work to get the visa application. And those are stressors. And they *do* stress you" [Interviewee #10; PMQ: India]. Physical effects included: sleep disturbance, memory issues, emotional eating leading to weight gain, fatigue, hair loss, somatic pains, and gut related symptoms. One IMG explained the ongoing stress of standing up for her work rights, even after completing the moratorium: "I'm good at setting boundaries but having to do it constantly… fighting for leave…and fighting for the right pay…I definitely felt burnt out…" [Interviewee #13; PMQ: UK]. Despite the struggles, interviewees also reported resilience and character building: "while I had that experience, it was horrible- but I don't regret it. I think it shaped the person that I am today. And I am very happy with the outcome" [Interviewee #32, Trinidad and Tobago].

**Triangulation of quantitative and qualitative data (see data in** S8 Table**).** Overall, there was high congruency between the quantitative and qualitative data. By triangulating results (S8 Table), we found that discrimination and disadvantage perceived by IMGs in the workplace was multifactorial and complex. For example, although we identified a high level of disadvantage reported by survey respondents (181/208; 87.0%), interview respondents explained that some of this disadvantage was not reflective of discrimination per se, but rather, an adjustment to a new workplace and culture or lack of established professional relationships as a newcomer. Survey data pointing directly to discriminatory experiences was congruent with clear descriptions of prejudice, stereotypes and discrimination in the qualitative data. For example, the quantitative data identified a substantial number of participants reported not being paid fairly for their level of work or qualification (55/117; 47%); supported by qualitative explanation of institutional paygrade structuring and allocation to senior positions due to experience but lack of equivalent financial reward; see S8 Table.

Statistically significant differences in the demographic data were supported by the data derived from interview participants. For example, qualitative results demonstrating the role of ethnicity and country of PMQ in the experiences of discrimination supported the statistical associations detected in quantitative data (p = 0.023 and p = 0.017 respectively). Similarly, native language as a statistically significant factor associating discrimination in the last five years (p = 0.047), was supported by qualitative data explaining accent as a trigger for discriminatory behaviour toward IMGs. Perpetrating sources of discrimination and personal discriminatory experiences identified in the quantitative data were more thoroughly explained by qualitative data, revealing workplace bullying by senior staff as a problem for many IMGs. Quantitative survey results identifying perceptions of discrimination effects on IMG career and health were supported by detailed explanations in the interview data; for example 68/205 (33.2%) survey respondents reported discrimination in the last five years affecting their career progression; explained by the interview data that this was due to multiple factors, such as a lack of stability from frequent relocations (due to job and visa requirements or bullying), unfair rostering of night shifts or rural allocations (impacting exam and training success), lack of prior experience recognition and a general preference for local graduates in training positions; see S8 Table.

## Discussion

The results of our study provide new insights into the personal experiences of IMGs in relation to inequitable treatment. The sequential explanatory mixed methods approach enriched our exploration by providing a broad understanding quantitatively, which was later further explained qualitatively, and strengthened in understanding by the triangulation process (S8 Table). Our study highlights the differences between disadvantage and discrimination perceived by IMGs in Australia, underlying factors and implications.

It is concerning that more than half of the survey respondents reported experiencing discrimination in the last five years. It is also pertinent to note that most reasons appeared to point toward racial biases (holding a foreign degree, language/accent and race/skin colour). This was supported by the statistically significant findings related to ethnicity, native language and country of PMQ. The British Medical Association's 2021 study into racism in medicine found that IMGs experienced racism more often than UK trained doctors and were twice as likely to consider racism as a barrier to career progression [34]. Our study supports these findings, and other reports identifying experiences of racism impacting IMG confidence, physical and mental wellbeing [11]. Furthermore, our results also confirm other studies which identify the compounding intersection of racial discrimination with religion and gender [34,35].

Indeed, although racism is largely considered unacceptable within the modern-day workplace code and conduct, workers may still be subject to subtle interpersonal exclusions, innocuous and sometimes unconscious manifestations of racially driven inequitable behaviours and microaggressions, manifesting as workplace bullying [36]. Workplace bullying is described as repeated unreasonable behaviours by a person/people toward a worker/s, creating a health and safety risk [37]. A 2024 national survey of trainee doctors in Australia found that 20% of IMG trainees and 38% of Aboriginal and/or Torres Strait Islander trainees had personally experienced bullying, harassment, discrimination and/or racism in the previous year, and that almost half (42% and 51% respectively) attributed the source as senior medical staff [38]. Our results support other international studies which report psychological distress and poorer job satisfaction in black and ethnic minority (BAME) doctors reporting bullying [34,35,39]. Other effects of bullying include burnout and increased accidents at work [40]. A surprising finding from our study was the implication of *both* IMG and non-IMG senior staff as perpetrators of bullying and discriminatory behaviour. IMG challenges in negotiating a new workplace

hierarchy have been previously reported [41]. However, the phenomenon of senior IMGs bullying other IMGs, is not described elsewhere in the literature to our knowledge. Theories postulated by our study participants (e.g., internal racism and cultural hierarchies) warrant further exploration in future research.

It can be argued that culturally diverse workforces are not necessarily indicative of inclusive workplace practices [42]. Ethnic minority healthcare staff are often over-represented in low wage positions and under-represented in leadership roles [42]. Indeed, IMGs working in Australia are overrepresented in rural and remote geographical locations [4]. This is particularly true for those working in General Practice where moratorium policies (e.g., restricted billing) may further compound inequities experienced, supported by our findings. System factors related to institutions and bureaucracy featured prominently in our study, as a source of both disadvantage and discrimination (S4 and S5 Figs). Most concerning was the exploitation perceived by interviewees, facilitated by current structural policies. Urgent investigation into such policies is warranted to reduce inequities. Furthermore, our study highlights the importance of educating IMGs of their workplace rights in Australia. Australian workplace laws protect the basic rights of all workers in Australia, irrespective of visa or citizenship status [43]. IMGs may contact The Fair Work Ombudsman to receive free advice about workplace concerns, or for education about workplace rights [44].

This research supports previously postulated recommendations to IMG inequity [11,42]. Long-term, strategic approaches from institutions are needed to embed change and foster a culture of shared responsibility and accountability toward inclusivity and fair practice [45,46]. Our results indicate that current institutional structures and policies in Australia which involve IMG assessment, regulatory licencing, employment, workplaces, training and immigration would benefit from review and change in order to ensure fairness and promotion of equitable practice.

Disadvantages experienced by IMGs as newcomers to a foreign work environment may be mitigated by experience sharing programs such as mentoring and peer support programs [11,42]. Such programs may assist by providing a safe forum for IMGs to openly discuss challenges navigating new workplaces, expectations and roles [11,42]. Other recommendations include practical workplace assistance, such as support for examinations, training and language/communication courses. This is particularly relevant for those IMGs located in isolated geographical regions where inequities to training support are greater [11]. Our results indicate that IMGs may benefit from communication skills courses to bridge written and spoken language deficits, thereby improving opportunities at interviews and examination performance; whilst also benefitting communication with other staff and patients. Institutions have a vital role in fostering an equitable and inclusive environment, thereby promoting a culture of workplace harmony [11]. The British Medical Association's Racism in Medicine report provides several examples of good workplace practice for promoting race equality and appreciation for cultural differences, for example, opportunities for safe and open discussions, supportive seniors, celebration of cultural events, actionable policies and anti-racism training [34,46]. Indeed, better understanding and acknowledgment of inequity affecting IMGs is required in order for productive change to ensue.

## Strengths and limitations

This research uncovers a controversial national health workforce issue, which has been neglected in the literature. It utilises participant voices to express personal experiences in a meaningful way. The sequential explanatory mixed methods approach, together with the process of triangulation (S8 Table), strengthened the identification of concepts and deep understanding of the issues at hand. Statistically significant results (Table 1) add strength

to data findings and promote interest for exploration in future studies. Despite the promising findings, our study is not without limitations. This study was exploratory in nature and not designed to detect a specific hypothesis with adequate statistical power. As such, some non-significant results may be due to insufficient power to detect true effects. Conversely, some significant findings could be the result of multiple comparisons. Additionally, the methods of recruitment (e.g., via social networks and snowball sampling) introduce potential bias to the study. Therefore, the convenience sample may limit the generalizability of the findings, as it may not fully represent the broader IMG population. Although the virtual interview format benefited our study by reducing participation bias, other biases may have been inadvertently introduced, e.g., missed non-verbal cues, an emphasis on first impressions etc [47].

Nonetheless, our study demonstrates that there is indeed a portion of IMGs in Australia who report discrimination and negative sequelae. Some important quantitative findings could not be fully explored in the interviews due to lack of opportunity within the interview sample, e.g., statistically significant association between reports of discrimination in the last five years and those not in full or part time work. Such findings lend an opening to future research in the area. Similarly, generalisations cannot be made from our study findings due to the lack of similar research in the area. However, our results provide a unique position to leverage other future studies exploring this important topic, both in Australia and internationally.

## Conclusion

Discrimination and disadvantage are reported by IMGs in Australia, with resultant detrimental effects on health and career. Institutions and workplaces have an important role in modifying current procedures and policies to reduce inequities, promote inclusion and better support IMGs. Further research and action in this space is urgently needed, as sustaining the IMG workforce is essential for national health protection.

## Supporting information

**S1 Appendix. Survey and interview questionnaires.**
(PDF)

**S2 Checklist. Good Reporting of a Mixed Methods Study.**
(PDF)

**S3 Table. Demographics of survey and interview participants.**
(PDF)

**S4 Figure. Selected reasons for IMGs reporting disadvantage.**
(PDF)

**S5 Figure. Perpetrating sources of discrimination within the last five years, as reported by affected respondents in sample.**
(PDF)

**S6 Table. Reports of discrimination.**
(PDF)

**S7 Figure. Self-reported effects of discrimination in the last five years.**
(PDF)

**S8 Table. Joint display of quantitative and qualitative data.**
(PDF)

## Author contributions

**Conceptualization:** Sunita Joann Rebecca Healey, Kristy Fakes, Bunmi Malau-Aduli, Balakrishnan R Nair.

**Data curation:** Sunita Joann Rebecca Healey.

**Formal analysis:** Sunita Joann Rebecca Healey, Kristy Fakes, Bunmi Malau-Aduli, Lucy Leigh.

**Funding acquisition:** Sunita Joann Rebecca Healey.

**Investigation:** Sunita Joann Rebecca Healey.

**Methodology:** Sunita Joann Rebecca Healey, Kristy Fakes, Bunmi Malau-Aduli.

**Project administration:** Sunita Joann Rebecca Healey, Balakrishnan R Nair.

**Supervision:** Kristy Fakes, Bunmi Malau-Aduli, Lucy Leigh, Balakrishnan R Nair.

**Validation:** Kristy Fakes, Bunmi Malau-Aduli, Lucy Leigh, Balakrishnan R Nair.

**Writing – original draft:** Sunita Joann Rebecca Healey.

**Writing – review & editing:** Sunita Joann Rebecca Healey, Kristy Fakes, Bunmi Malau-Aduli, Lucy Leigh, Balakrishnan R Nair.

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
