## [Decision Letter · Decision Letter 0]

7 Nov 2024

PONE-D-24-43854Exploring experiences of work-related inequitable treatment among international medical graduates (IMGs): a sequential explanatory mixed methods studyPLOS ONE

Dear Dr. Healey,

Thank you for submitting your manuscript to PLOS ONE. After careful consideration, we feel that it has merit but does not fully meet PLOS ONE’s publication criteria as it currently stands. Therefore, we invite you to submit a revised version of the manuscript that addresses the points raised during the review process.

**ACADEMIC EDITOR: ** **Please address the reviewer comments as under and return for further consideration for submission** 

We look forward to receiving your revised manuscript.

Kind regards,

Souparno Mitra, M.D.

Academic Editor

PLOS ONE

Journal Requirements:

2. Thank you for stating the following in the Competing Interests section: SJRH and BRN are both IMGs and work closely with IMGs, within the Workplace Based Assessment Program, Hunter New England Health. Other authors (KF, BMA, LL) do not demonstrate any competing interests. 

Reviewers' comments:

Reviewer's Responses to Questions

**Comments to the Author**

1. Is the manuscript technically sound, and do the data support the conclusions?

Reviewer #1: Yes

Reviewer #2: Yes

2. Has the statistical analysis been performed appropriately and rigorously? 

Reviewer #1: Yes

Reviewer #2: Yes

3. Have the authors made all data underlying the findings in their manuscript fully available?

Reviewer #1: Yes

Reviewer #2: Yes

4. Is the manuscript presented in an intelligible fashion and written in standard English?

Reviewer #1: Yes

Reviewer #2: Yes

5. Review Comments to the Author

Reviewer #1: disadvantage is different from discrimination explain how these two terms differ in your study.

Be more clear in discussion on using social networks and snowball sampling may not fully represent all IMGs.

Add a short reflection that why senior IMGs bully others could be cultural hierarchies, internalized discrimination.

your recommendations are too general be more specific like mentoring program , support for exam prepartion.

page 18 about cultural behaviors please simplify complex sentences they too long difficult to follow.

And make sure every figure and table is mentioned where it is relevant in the text.

Reviewer #2: This study highlights the widespread inequities and discrimination faced by International Medical Graduates (IMGs) in Australia, particularly emphasizing the effects of racial biases related to foreign degrees, language, and ethnicity. The sample size is small, which limits the generalizability of the findings.The use of both quantitative and qualitative methods enhanced the analysis, revealing important patterns and encouraging further investigation in future studies. While the results are promising, the exploratory nature of the research and the use of a convenience sample present limitations. Given the lack of comparable studies in this area, these findings provide a foundation for future research, promoting a deeper understanding of these issues.

6. PLOS authors have the option to publish the peer review history of their article (what does this mean? ). If published, this will include your full peer review and any attached files.

**Do you want your identity to be public for this peer review?** For information about this choice, including consent withdrawal, please see our Privacy Policy .

Reviewer #1: No

Reviewer #2: No

---

## [Author Response · Author response to Decision Letter 1]

15 Nov 2024

Many thanks for taking the time to review this re-submission, alongside the Response to Reviewer document, detailing minor revisions as requested. Kind regards, Dr Healey

---

## [Decision Letter · Decision Letter 1]

10 Jan 2025

PONE-D-24-43854R1Exploring experiences of work-related inequitable treatment among international medical graduates (IMGs): a sequential explanatory mixed methods studyPLOS ONE

Dear Dr. Healey,

Thank you for submitting your manuscript to PLOS ONE. After careful consideration, we feel that it has merit but does not fully meet PLOS ONE’s publication criteria as it currently stands. Therefore, we invite you to submit a revised version of the manuscript that addresses the points raised during the review process.

**ACADEMIC EDITOR: ** **Please address reviewer comments in order to consider further for acceptance. **

We look forward to receiving your revised manuscript.

Kind regards,

Souparno Mitra, M.D.

Academic Editor

PLOS ONE

Journal Requirements:

Reviewers' comments:

Reviewer's Responses to Questions

**Comments to the Author**

1. If the authors have adequately addressed your comments raised in a previous round of review and you feel that this manuscript is now acceptable for publication, you may indicate that here to bypass the “Comments to the Author” section, enter your conflict of interest statement in the “Confidential to Editor” section, and submit your "Accept" recommendation.

Reviewer #3: All comments have been addressed

Reviewer #4: All comments have been addressed

Reviewer #5: (No Response)

2. Is the manuscript technically sound, and do the data support the conclusions?

Reviewer #3: Yes

Reviewer #4: Yes

Reviewer #5: Yes

3. Has the statistical analysis been performed appropriately and rigorously? 

Reviewer #3: I Don't Know

Reviewer #4: I Don't Know

Reviewer #5: Yes

4. Have the authors made all data underlying the findings in their manuscript fully available?

Reviewer #3: Yes

Reviewer #4: Yes

Reviewer #5: Yes

5. Is the manuscript presented in an intelligible fashion and written in standard English?

Reviewer #3: Yes

Reviewer #4: Yes

Reviewer #5: Yes

6. Review Comments to the Author

Reviewer #3: (No Response)

Reviewer #4: Overall this is a very well-written article. It provides an excellent case for the need for a new study and research question, where there is a paucity of research. Abstract was clearly presented, with pertinent and was easy to read. The introduction aptly discussed prior research in this topic and briefly mentioned their results. It also describes the importance of IMGs to foreign economies, and the unique challenges they face to fulfill that need. Methods have been well-discussed, with good use of tables and figures. Authors can consider placing some tables and figures in the body of the article rather than supplemental material, as it may make it easier to follow visually. Results and discussion section is thorough and describes the nuances of the results. It also discusses the biases that may be present in the survey as compared to the interview method used. Limitations and strengths of the study are well-described. Compliments to the authors for presenting their information in a clear, concise way that is easy to read and is thought-provoking for further research.

Reviewer #5: Extremely informative article, and nice to see more studies popping up about IMG disparities in multiple countries. Please see attached PDF file with all edits and comments for the article.

7. PLOS authors have the option to publish the peer review history of their article (what does this mean? ). If published, this will include your full peer review and any attached files.

**Do you want your identity to be public for this peer review?** For information about this choice, including consent withdrawal, please see our Privacy Policy .

Reviewer #3: **Yes: ** Nikhil Tondehal

Reviewer #4: No

Reviewer #5: **Yes: ** Arun Prasad

---

## [Author Response · Author response to Decision Letter 2]

13 Jan 2025

Please see response to reviewers document attached

---

## [Decision Letter · Decision Letter 2]

30 Jan 2025

Exploring experiences of work-related inequitable treatment among international medical graduates (IMGs): a sequential explanatory mixed methods study

PONE-D-24-43854R2

Dear Dr. Healey,

We’re pleased to inform you that your manuscript has been judged scientifically suitable for publication and will be formally accepted for publication once it meets all outstanding technical requirements.

Kind regards,

Souparno Mitra, M.D.

Academic Editor

PLOS ONE

Additional Editor Comments (optional):

Reviewers' comments:

Reviewer's Responses to Questions

**Comments to the Author**

1. If the authors have adequately addressed your comments raised in a previous round of review and you feel that this manuscript is now acceptable for publication, you may indicate that here to bypass the “Comments to the Author” section, enter your conflict of interest statement in the “Confidential to Editor” section, and submit your "Accept" recommendation.

Reviewer #3: All comments have been addressed

Reviewer #5: All comments have been addressed

2. Is the manuscript technically sound, and do the data support the conclusions?

Reviewer #3: Yes

Reviewer #5: Yes

3. Has the statistical analysis been performed appropriately and rigorously? 

Reviewer #3: Yes

Reviewer #5: Yes

4. Have the authors made all data underlying the findings in their manuscript fully available?

Reviewer #3: Yes

Reviewer #5: Yes

5. Is the manuscript presented in an intelligible fashion and written in standard English?

Reviewer #3: Yes

Reviewer #5: Yes

6. Review Comments to the Author

Reviewer #3: (No Response)

Reviewer #5: Thank you for making the edits, it looks good - and an extremely important for topic for IMGs worldwide. Best of luck in getting it published!

7. PLOS authors have the option to publish the peer review history of their article (what does this mean? ). If published, this will include your full peer review and any attached files.

**Do you want your identity to be public for this peer review?** For information about this choice, including consent withdrawal, please see our Privacy Policy .

Reviewer #3: **Yes: ** Nikhil Tondehal

Reviewer #5: **Yes: ** Arun Prasad

---

## [Editor Report · Acceptance letter]

PONE-D-24-43854R2

PLOS ONE

Dear Dr. Healey,

I'm pleased to inform you that your manuscript has been deemed suitable for publication in PLOS ONE. Congratulations! Your manuscript is now being handed over to our production team.

Kind regards,

on behalf of

Dr. Souparno Mitra

Academic Editor

PLOS ONE